# The glyoxylate shunt is essential for desiccation tolerance in *C. elegans* and budding yeast

Cihan Erkut[1][†], Vamshidhar R Gade[1], Sunil Laxman[2], Teymuras V Kurzchalia[1]*

[1]Max Planck Institute of Molecular Cell Biology and Genetics, Dresden, Germany; [2]Institute for Stem Cell Biology and Regenerative Medicine, Bangalore, India

**Abstract** Many organisms, including species from all kingdoms of life, can survive desiccation by entering a state with no detectable metabolism. To survive, *C. elegans* dauer larvae and stationary phase *S. cerevisiae* require elevated amounts of the disaccharide trehalose. We found that dauer larvae and stationary phase yeast switched into a gluconeogenic mode in which metabolism was reoriented toward production of sugars from non-carbohydrate sources. This mode depended on full activity of the glyoxylate shunt (GS), which enables synthesis of trehalose from acetate. The GS was especially critical during preparation of worms for harsh desiccation (preconditioning) and during the entry of yeast into stationary phase. Loss of the GS dramatically decreased desiccation tolerance in both organisms. Our results reveal a novel physiological role for the GS and elucidate a conserved metabolic rewiring that confers desiccation tolerance on organisms as diverse as worm and yeast.

*For correspondence: kurzchalia@mpi-cbg.de

Present address: [†]Structural and Computational Biology Unit, European Molecular Biology Laboratory, Heidelberg, Germany

**Competing interests:** The authors declare that no competing interests exist.

## Introduction

Terrestrial organisms regularly encounter severe drought. For species with no means of preventing evaporative water loss, drought might result in desiccation, and eventually death. To cope with this environmental insult, many organisms enter an ametabolic state known as anhydrobiosis (*Keilin, 1959*; *Leprince and Buitink, 2015*). In this state, organisms can persist in the absence of water for a long period of time; when water becomes available, they exit the anhydrobiotic state and fully resume their normal activities. The nematode *Caenorhabditis elegans* and the budding yeast *Saccharomyces cerevisiae* are excellent anhydrobiotes. Studies of these two model organisms have revealed various strategies for desiccation tolerance, many of which appear to be broadly conserved among other anhydrobiotes (*Dupont et al., 2014*; *Erkut and Kurzchalia, 2015*).

One strategy for anhydrobiosis common to both worm and yeast is the biosynthesis and accumulation of trehalose (*Erkut et al., 2011*; *Tapia and Koshland, 2014*), a disaccharide made of two alpha-linked glucose moieties (*Elbein, 2003*). In *C. elegans,* trehalose preserves the native packing of membranes in the dried state (*Erkut et al., 2011*; *2012*) and stabilizes membranes against the adverse effects of fast rehydration (*Abusharkh et al., 2014*). In yeast, trehalose also functions as a long-lived chaperone, preventing protein aggregation upon desiccation (*Tapia and Koshland, 2014*). These observations suggest that this disaccharide plays conserved roles in desiccation tolerance. However, the metabolic basis for synthesis of trehalose remains largely unknown. In this study, we sought to identify the source of trehalose carbons and the pathway(s) that promote trehalose biosynthesis and accumulation.

Neither *C. elegans* nor *S. cerevisiae* invests in trehalose production during growth and development. By contrast, in their non-proliferative stages, i.e., the dauer larva in *C. elegans* (as shown in *Penkov et al., 2015*, and this study) and stationary phase in yeast (*François and Parrou, 2001*;

**eLife digest** Many organisms can survive losing all the water from their body in periods of severe drought by suspending their life. This ability is called anhydrobiosis (from the Greek for 'life without water'). When the desiccated organisms encounter water again, they resume life as normal. Two organisms commonly used in research, a roundworm called *Caenorhabditis elegans* and a yeast called *Saccharomyces cerevisiae*, are anhydrobiotes.

To survive without water, anhydrobiotes alter the chemical reactions that sustain their life, and so change their metabolic state. The organisms also produce molecules that preserve the structure of their cells. One such essential molecule is a sugar called trehalose. However, both worms and yeast can only enter anhydrobiosis during particular stages of life where they do not eat. So where does the trehalose come from?

Erkut et al. have now addressed this question by studying the metabolism of *C. elegans* and *S. cerevisiae* as these species entered anhydrobiosis. The experiments revealed that while preparing for desiccation, both species change their metabolism to favor creating sugars rather than releasing energy. In this process, the worms and yeast use a biochemical pathway called the glyoxylate shunt, which can convert fat or acetic acid into sugar. Genetic mutations that deactivate this pathway severely reduce the ability of both organisms to produce trehalose and tolerate desiccation. From these findings, Erkut et al. conclude that the source of trehalose in non-feeding worms is their fat deposits, while in yeast it is acetate: a molecule that is derived from ethanol, the end-product of the fermentation process.

The glyoxylate shunt had been thought only to be a non-essential biochemical shortcut of another well-known metabolic pathway called the Krebs cycle. Now that Erkut et al. have shown that the glyoxylate shunt has its own specific biological role, further investigation is needed to understand how it is activated to act as a metabolic switch. The molecules that regulate similar metabolic transitions will also need to be identified in future studies. Ultimately, understanding these processes could present new ways of diagnosing and treating metabolic diseases such as diabetes and cancer.

*Werner-Washburne et al., 1993*), both species devote a substantial amount of their internal carbon reserve to trehalose biosynthesis. In the non-feeding dauer larva (the only desiccation-tolerant stage of the *C. elegans* life cycle), trehalose levels rise dramatically upon exposure to mild desiccation stress (preconditioning) (*Erkut et al., 2011*). Similarly, stationary phase yeast cells, which are tolerant to desiccation, also accumulate trehalose (*Calahan et al., 2011*). Thus, in these specific developmental stages, these organisms must be able to divert available carbon sources to the production of sugars.

In these non-growing stages, both worm and yeast must enter metabolic modes distinct from those that are active during growth. Reproductive stage larvae of *C. elegans* feed on bacteria, from which they ingest mostly lipids and proteins, and to some extent sugars; they assimilate these nutrients via glycolysis and/or the TCA cycle to produce energy (*Figure 1A*, *Figure 1—figure supplement 1A*). On the other hand, when fed its preferred carbon source (glucose), budding yeast grows exponentially and uses fermentative glycolysis for its energetic and biosynthetic needs (*Figure 1B*). Under these conditions, the cells secrete ethanol as well as acetate.

Both species shift their metabolism during the transition to non-growing stages. The dauer larva relies on its internal carbon reserves, which are mainly triacylglycerols (TAGs) (*Hellerer et al., 2007*; *Narbonne and Roy, 2008*), but must also retain the ability to produce sugars. In yeast, as glucose is consumed and glucose concentrations drop, cells undergo the diauxic shift, i.e., a transition to respiratory metabolism (*Schweizer and Dickinson, 2004*). Following this shift, yeast produce acetyl-CoA from accumulated ethanol, acetate, and glycerol, and switch to oxidative phosphorylation via the TCA cycle (*Schweizer and Dickinson, 2004*), which provides energy as well as precursor metabolites for amino acid biosynthesis and gluconeogenesis. Finally, in the stationary phase, yeast cells accumulate trehalose and glycogen (*François and Parrou, 2001*; *Schweizer and Dickinson, 2004*; *Werner-Washburne et al., 1993*).

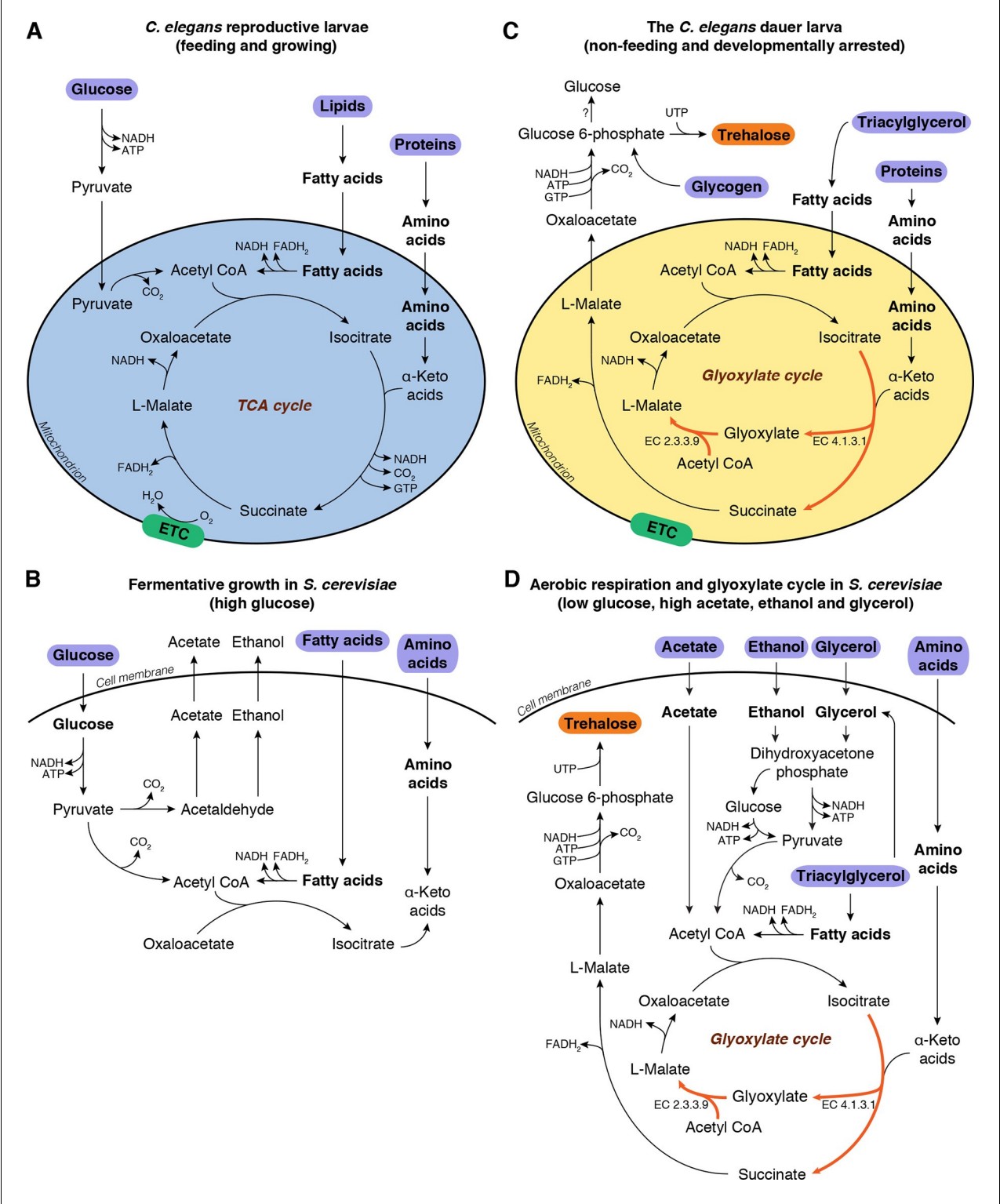

**Figure 1.** Metabolic modes of *C. elegans* and *S. cerevisiae*. (A) *C. elegans* reproductive larvae, which are feeding and growing, can utilize nutrients (purple) via TCA cycle and produce energy. Mitochondria are in a catabolic mode (blue). (B) During fermentative growth, *S. cerevisiae* uses glucose to produce energy via glycolysis. (C) The non-feeding dauer larva utilizes internal TAG reserves via GS to drive gluconeogenesis and produce trehalose (orange). Mitochondria are in an anabolic mode (yellow). (D) In low glucose, high acetate, ethanol and glycerol regime, yeast switches to gluconeogenesis via GS.

*Figure 1 continued on next page*

*Figure 1 continued*

The following figure supplement is available for figure 1:

**Figure supplement 1.** Metabolic pathways of glycolysis, gluconeogenesis, TCA cycle and glyoxylate shunt reactions during preconditioning.

These observations imply that in order to synthesize trehalose, both organisms must undergo a transition to a gluconeogenic mode in which they synthesize glucose or glucose-6-phosphate from non-carbohydrate precursors. How is this transition implemented? In theory, the TCA cycle could provide the intermediates required for gluconeogenesis, but this pathway generates substantial amounts of ATP, as well as NADH, which must be oxidized to NAD$^+$ to maintain cellular redox balance (*Voet and Voet, 2010*). At first glance it seems counterintuitive that these two processes run in parallel, considering the low energetic demands of dauer larvae and stationary phase yeast cells. However, cells could be driven into gluconeogenesis via an alternate route, the glyoxylate shunt (GS) (*Figure 1C,D*, depicted in red). The GS has been implicated in anhydrobiosis in the nematode *Aphelencus avenae* (*Madin et al., 1985*). We hypothesized that, in *C. elegans* dauer larvae and stationary phase yeast cells, the GS serves a critical function in anabolic processes required for desiccation tolerance, in particular by enabling or promoting gluconeogenesis for trehalose biosynthesis.

The GS, a shortcut in the TCA cycle (*Kornberg and Madsen, 1958*), is conserved in bacteria (*Kornberg, 1966*), fungi (*Lopez-Boado et al., 1988*; *Lorenz and Fink, 2001*), protists (*Levy and Scherbaum, 1965*; *Nakazawa et al., 2005*), nematodes (*Liu et al., 1995*; *Madin et al., 1985*; *Siddiqui et al., 2000*), and plants (*Eastmond and Graham, 2001*; *Kornberg and Beevers, 1957*). It bypasses two $CO_2$-releasing steps of the TCA cycle (*Figure 1C*, *Figure 1—figure supplement 1A*, reactions 3 and 4) to produce succinate, and incorporates an additional molecule of acetyl-CoA to form L-malate from glyoxylate (*Figure 1C,D*, *Figure 1—figure supplement 1A*, reaction 10). Instead of remaining within the TCA cycle, excess malate can be converted into oxaloacetate and diverted into gluconeogenesis (*Figure 1C*, *Figure 1—figure supplement 1A*, reaction 12) (*Voet and Voet, 2010*). Thus, the GS serves as a prototypical anaplerotic pathway, leading to the accumulation of critical TCA cycle intermediates, particularly oxaloacetate, which can be consumed for gluconeogenesis. Moreover, this pathway generates less ATP and NADH than the TCA cycle (*Kornberg, 1966*).

To date, the biological importance of the GS has been largely ignored, and its physiological functions remain obscure. The GS has primarily been studied in the context of microbial sporulation and growth (*Kornberg and Krebs, 1957*; *Megraw and Beers, 1964*), fungal virulence (*Lorenz and Fink, 2001*), and plant seed germination (*Eastmond et al., 2000*). However, the GS is not physiologically essential to any of these processes (*Voet and Voet, 2010*). On the other hand, it is astonishing that *C. elegans*, a nematode and thus a member of the animal kingdom, has the full set of enzymes required for the GS (*Liu et al., 1995*). Although the GS has been proposed to be involved in sugar homeostasis in the worm (*Frazier and Roth, 2009*), its absence results in neither a detectable phenotype nor any effect on wild-type adult lifespan. However, it may be required for the extended longevity of some mitochondrial mutants (*Gallo et al., 2011*). Thus, at present, no physiological role has been definitively assigned to this pathway in the worm.

Here, we present evidence that the dauer larva is in a hypoaerobic, gluconeogenic state, which enables efficient production of trehalose using internal reserves (TAGs and amino acids). Importantly, during preconditioning, the GS is the major pathway for conversion of TAGs into trehalose; in its absence, the dauer larva cannot produce sufficient trehalose to survive desiccation. Expanding our studies to the budding yeast, we discovered that *S. cerevisiae* utilizes a similar metabolic strategy, relying on the GS to drive trehalose synthesis and achieve desiccation tolerance. These results reveal, for the first time, a functionally conserved and central role for the GS in a process that is essential for survival under certain conditions.

## Results

### The dauer larva is in a hypoaerobic, gluconeogenic mode

We characterized the energetic/metabolic states of the dauer larva and its parallel reproductive stage (the L3 larva). Dauer larvae are metabolically less active than L3 larvae (*Burnell et al., 2005*;

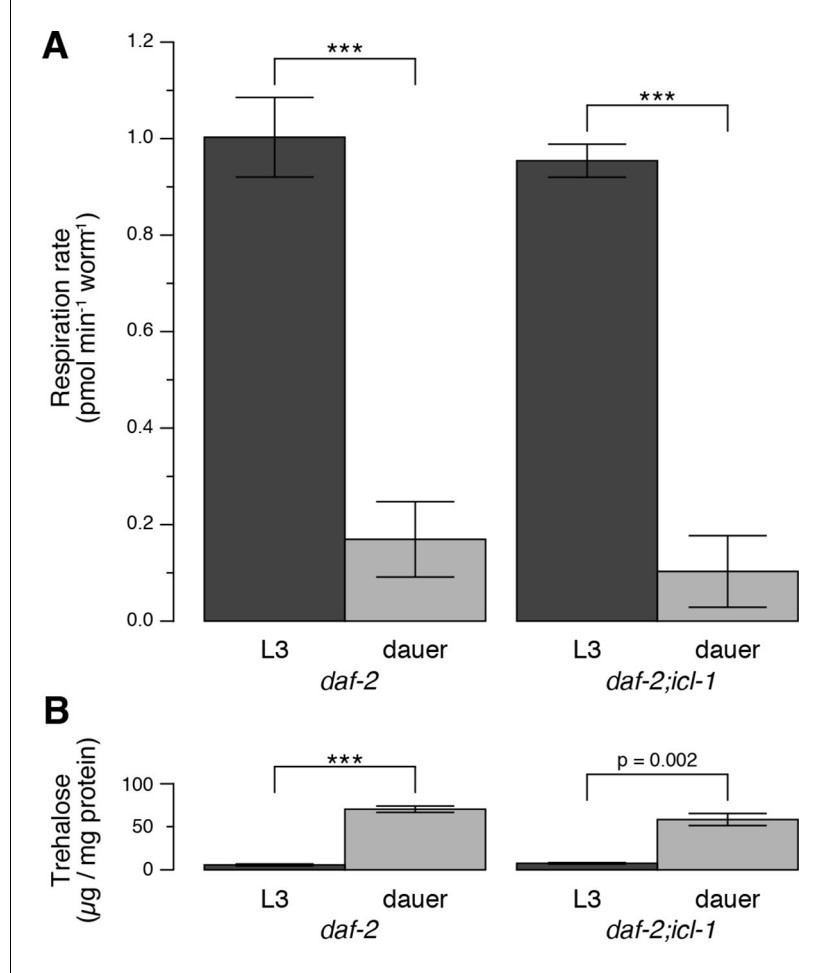

**Figure 2.** Energetic modes of *C. elegans* reproductive and dauer larvae. (**A**) Respiration rates in terms of OCR difference between water-treated and azide-treated worms (n = 4 for each group). ANOVA shows that in both strains, L3 larvae consume significantly more oxygen than dauer larvae ($F_{1,12}$ = 1469, p < 0.001). There is also a minor effect of strain on oxygen consumption ($F_{1,12}$ = 6.864, p = 0.022), however there is no interaction between the larval stage and the strain ($F_{1,12}$ = 0.166, p = 0.691). Error bars show 95% confidence intervals. (**B**) Steady-state trehalose levels of *daf-2* and *daf-2;icl-1*, L3 and dauer larvae (n = 3 for each group). L3 larvae produce less trehalose than dauer larvae ($F_{1,8}$ = 92.814, p < 0.001) independent of the strain ($F_{1,8}$ = 0.083, p = 0.781). Error bars show standard error of the mean. *p < 0.001.

The following figure supplement is available for figure 2:

**Figure supplement 1.** Details of oxygen consumption rate measurements.

---

*Kimura et al., 1997*; *O'Riordan and Burnell, 1989*; *1990*; *Vanfleteren and DeVreese, 1996*). As an indicator of metabolic activity, we compared the respiration rates of dauer and L3 larvae. To obtain large quantities of homogeneous L3 or dauer larva populations, we used the temperature-sensitive dauer-constitutive *daf-2(e1370)* strain. Oxygen consumption rates (OCRs) in these larvae were measured using an extracellular flux analyzer (*Figure 2—figure supplement 1*). To measure mitochondrial OCR, we specifically inhibited Complex IV with sodium azide (*Figure 2—figure supplement 1*) and calculated the respiration rate as the difference between the overall OCRs of water- and azide-treated worms (*Figure 2A*). In a given concentration of environmental oxygen, mitochondria of dauer larvae consumed ~5-fold less oxygen than those of L3 larvae (*Figure 2A*), indicating that dauer larvae exist in a hypoaerobic mode. Moreover, dauer larvae contain much less ATP than L3

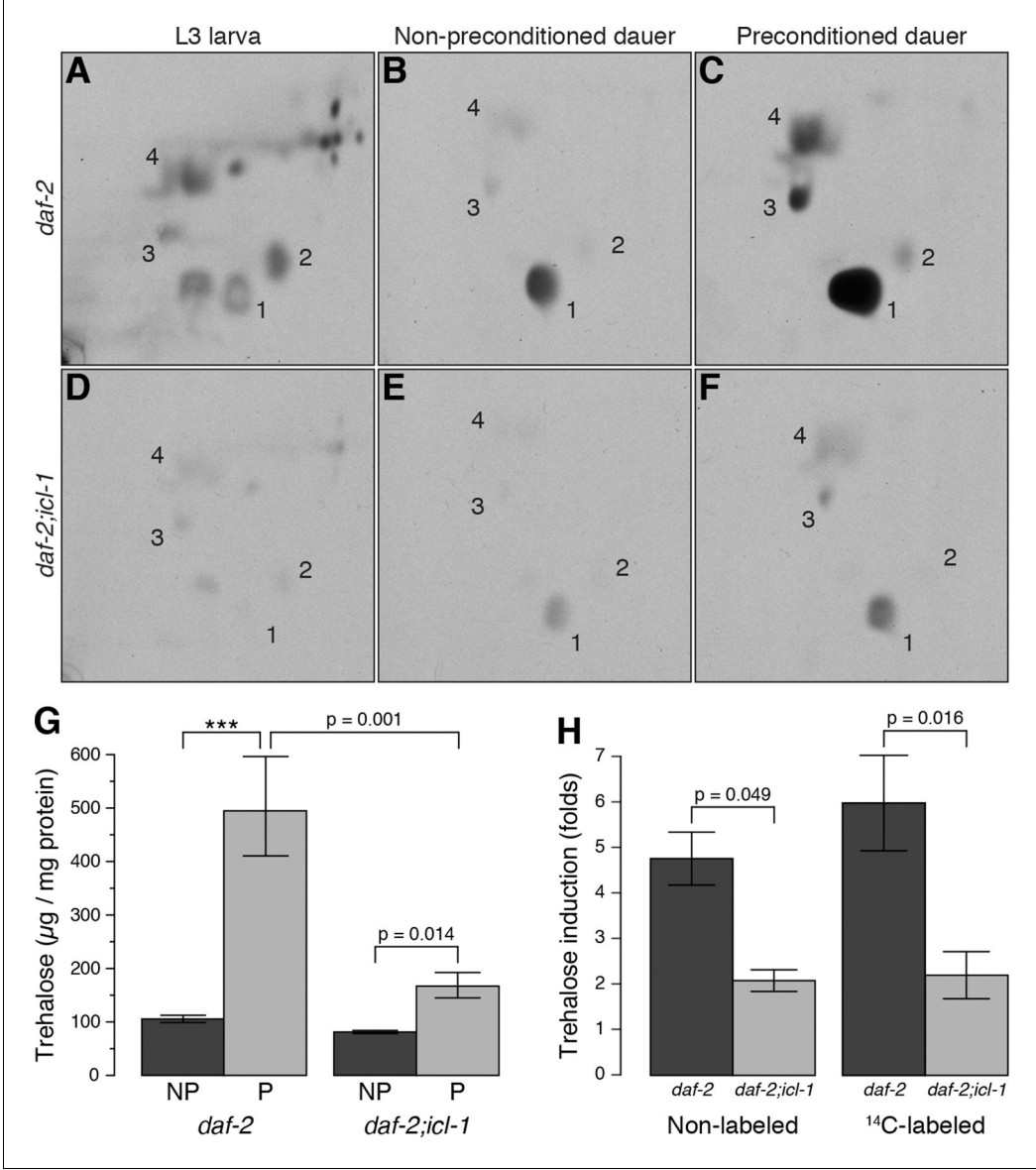

**Figure 3.** Metabolic modes of *C. elegans* reproductive and dauer larvae. (**A–C**) Radioactively labeled metabolites of *daf-2* L3, as well as non-preconditioned (NP) and preconditioned (P) dauer larvae. Enumerated spots indicate trehalose (1), glucose (2), glutamate (3) and glutamine (4). (**D–F**) The same analysis for *daf-2;icl-1*. Equivalent metabolome extracts were separated and exposed for 2 days for both strains and larval/experimental conditions. (**G**) Steady state trehalose levels before and after preconditioning in *daf-2* and *daf-2;icl-1* dauer larvae (n = 3 for each group). Both strains elevate their trehalose levels upon preconditioning (ANOVA for preconditioning reports $F_{1,8}$ = 85.20, p < 0.001) but to different extents (ANOVA for strain reports $F_{1,8}$ = 30.11, p < 0.001; interaction between strain and preconditioning $F_{1,8}$ = 11.26, p = 0.010). Error bars show standard error of the mean. *p < 0.001. (**H**) Induction of non-labeled and $^{14}$C-labeled trehalose upon preconditioning in *daf-2* and *daf-2;icl-1* dauer larvae expressed as fold changes (n = 3 for each group). ANOVA shows that *daf-2* larvae induce both non-labeled and labeled trehalose more than *daf-2;icl-1* larvae ($F_{1,8}$ = 26.229, p < 0.001) however induction in labeled trehalose does not differ from non-labeled ($F_{1,8}$ = 0.343, p = 0.571). Error bars show standard error of the mean.

The following figure supplements are available for figure 3:

**Figure supplement 1.** Details for the detection of metabolites.

**Figure supplement 2.** Trehalose 6-phosphate synthase (TPS) levels in worm and yeast.

larvae (*Penkov et al., 2015*; *Wadsworth and Riddle, 1989*), indicating that they are also hypometabolic.

Next, we compared trehalose levels in L3 vs. non-feeding dauer larvae. The latter accumulated substantially larger amounts of trehalose (*Figure 2B*). This observation suggested that, in addition to being hypometabolic, dauer larvae rearrange their metabolism to favor intensive gluconeogenesis, leading to trehalose accumulation.

To investigate this possibility, we adopted an approach that combined metabolic labeling with 2-dimensional high-performance thin-layer chromatography (2D-HPTLC). This relatively simple method enabled us to detect major small-molecules, including amino acids, sugars, and intermediates of the TCA cycle (*Figure 3—figure supplement 1A*). First, we labeled *C. elegans* metabolites by feeding the worms $^{14}$C-acetate-supplemented bacteria until they formed L3 or dauer larvae. This labeling strategy allowed us to detect and identify metabolites derived from $^{14}$C-acetate that has entered the TCA cycle. Subsequently, we extracted the metabolites from worms, separated the extracts into organic and aqueous phases, and analyzed the latter with 2D-HPTLC.

The aqueous phase of L3 extract contained many labeled compounds, including various amino acids (*Figure 3A*, *Figure 3—figure supplement 1A*), but trehalose was not abundant (*Figure 3A*, spot 1). Thus, in this growth stage, the TCA cycle is mainly cataplerotic: in addition to reducing NAD$^+$ and producing ATP, L3 larvae use intermediates to synthesize various building blocks such as amino acids, nucleotides, and sugars. By contrast, the aqueous fraction of dauer larvae contained only a limited number of metabolites, and fluorograms of this extract had one predominant spot, i.e., trehalose (*Figure 3B*, spot 1). Other, barely detectable spots corresponded to glucose, glutamate, and glutamine (*Figure 3B*, spots 2, 3, and 4, respectively). These data suggest that metabolism in dauer larvae is almost entirely switched to a gluconeogenic mode in which sugars are produced by non-carbohydrate sources (acetate/fatty acids).

Previously, we showed that preconditioning of the dauer larva prior to harsh desiccation induces production of a massive amount of trehalose (*Erkut et al., 2011*). For preconditioning, worms are treated with mild desiccation at 98% relative humidity (RH) for an extended period of time (4 days), after which they can survive in the almost complete absence of water (*Erkut et al., 2011*). In this study, we preconditioned worms in this manner, and then analyzed the organic and aqueous fractions of radioactively labeled dauer larvae before and after preconditioning. In the organic phase, the amount of radioactivity incorporated into TAGs decreased substantially during preconditioning (*Figure 3—figure supplement 1B*). At the same time, preconditioning dramatically increased the level of radioactively labeled trehalose (*Figure 3C*, spot 1), and the amounts of glutamate and glutamine also increased (*Figure 3C*, spots 3 and 4, respectively; this observation is discussed later). These results suggest that the dauer larva takes advantage of its gluconeogenic mode to boost trehalose synthesis upon desiccation stress.

We next asked how the transition to this gluconeogenic mode is reflected in the transcriptome. Previously, we surveyed differential expression of *C. elegans* genes during preconditioning (*Erkut et al., 2013*). In this study, we revisited our data to focus on genes involved in the TCA cycle and gluconeogenesis (*Figure 1—figure supplement 1A*). Transcripts encoding enzymes required for gluconeogenesis, *mdh-1*, *mdh-2*, and *pck-2* (*Figure 1—figure supplement 1B*, enzymes 9 and 12), were expressed at relatively high levels in dauer larvae even before preconditioning. Moreover, *mdh-1* (cytosolic malate dehydrogenase) and *pck-2* (phosphoenolpyruvate carboxykinase), both of which are crucial for gluconeogenesis, were significantly upregulated during preconditioning, consistent with the increase in gluconeogenesis and sugar accumulation observed in the dauer larva.

Collectively, our data demonstrate that, during dauer formation, worms enter a gluconeogenic mode associated with a large increase in the levels of sugars such as trehalose, and that this phenomenon is even more pronounced during preconditioning.

## An intact GS is required for utilization of acetate/fatty acids for trehalose biosynthesis

We hypothesized that the GS in dauer larvae plays an important role in gluconeogenesis, and thus in trehalose biosynthesis. We tested this idea in worm lines having no functional GS.

In plants, yeast, and bacteria, two enzymes are responsible for the GS: isocitrate lyase (EC 4.1.3.1), which breaks isocitrate down to glyoxylate and succinate, and malate synthase (EC 2.3.3.9), which condenses glyoxylate and acetyl-CoA to produce L-malate (*Figure 1C*, *Figure 1—*

*figure supplement 1A*) (*Cozzone, 1998*). In *C. elegans*, these two enzymes are combined in one protein, ICL-1 (formerly known as GEI-7), which has both isocitrate lyase and malate synthase domains, and can thus carry out both reactions (*Liu et al., 1995*). We produced a strain (*daf-2; icl-1*) with a deletion mutation in *icl-1*, and then exploited the *daf-2* background to produce large populations of pure dauer larvae. The deletion in the *icl-1(ok531)* allele introduces a frame-shift and an early stop codon (A373*), which should completely inactivate the GS strains harboring this mutation.

Compared to *daf-2*, the dauer and L3 larvae of *daf-2;icl-1* exhibited no difference in respiration rate (*Figure 2A*) or basal trehalose levels (*Figure 2B*), indicating that the GS has no influence on oxygen consumption or basal gluconeogenesis. By contrast, trehalose induction upon preconditioning differed dramatically between *daf-2* and *daf-2;icl-1* dauer larvae. We quantitated the total amount of trehalose, normalized to the amount of soluble protein, in both strains before and after preconditioning (*Figure 3G*). Similar to our previous findings (*Erkut et al., 2011*), the total trehalose level in *daf-2* dauer larvae was ~ 100 µg trehalose/mg protein at baseline, and increased ~ 5-fold upon preconditioning (*Figure 3G,H*). In *daf-2;icl-1*, although the initial level of trehalose was the same as that of *daf-2*, the increase was only 2-fold (*Figure 3G,H*). This suggests that the major source of trehalose during preconditioning is the GS.

To further investigate this possibility, we labeled *daf-2;icl-1* larvae with $^{14}$C-Ac, as described above for *daf-2* (*Figure 3D–F*). Incorporation of radioactivity into trehalose was considerably reduced in *daf-2;icl-1* relative to that in *daf-2* (compare *Figure 3B and E*). Nevertheless, as in *daf-2*, the level of $^{14}$C-labeled trehalose increased in *daf-2;icl-1* (*Figure 3E,F*). Densitometry of fluorogram spots revealed that, during preconditioning, radioactively labeled trehalose increased ~ 6-fold in *daf-2*, but only 2-fold in *daf-2;icl-1* (*Figure 3H*). The average increase in labeled trehalose in *daf-2* was larger than the increase in total (i.e., unlabeled) trehalose (*Figure 3H*), suggesting preferential use of lipid sources for sugar production in this strain. By contrast, in *daf-2;icl-1*, the levels of total and labeled trehalose increased to similar extents. These differences in trehalose induction levels between *daf-2* and *daf-2;icl-1* cannot be assigned to the trehalose biosynthetic pathway because the trehalose 6-phosphate synthase (*Figure 1—figure supplement 1A*, reaction 25) activity of *daf-2* is not higher than that of *daf-2;icl-1* (*Figure 3—figure supplement 2A*). Taken together, these results suggest that the utilization of acetate (and thus fatty acids) for gluconeogenesis and trehalose biosynthesis depends on the existence of a functional GS.

## The GS is essential for desiccation tolerance of the dauer larva

Next, we asked whether the absence of the GS affects desiccation tolerance. For this purpose, we determined the survival rates of dauer larvae after mild (98% RH) and harsh (60% RH) desiccation. As described above and in our previous studies (*Erkut et al., 2011*), preconditioning induced trehalose accumulation strongly in *daf-2* and slightly in *daf-2;icl-1* (*Figure 3G*). As expected, a strain harboring a knockout of trehalose 6-phosphate synthase (*daf-2;ΔΔtps*) was unable to synthesize trehalose (*Figure 4A*). The desiccation survival assay revealed that all strains were equally tolerant to mild desiccation at 98% RH (*Figure 4B*). However, *daf-2;icl-1* was much more sensitive to harsh desiccation than *daf-2*, and exhibited very poor survival under those conditions (*Figure 4B*), although it was significantly more tolerant than *daf-2;ΔΔtps* (*Figure 4B*). These results suggest that a threshold level of trehalose must be reached during preconditioning in order for the worm to survive harsh desiccation.

To investigate the possibility that the GS might play a role in protection against environmental insults unrelated to desiccation stress, we also challenged dauer larvae with heat shock. Both *daf-2* and *daf-2;icl-1* dauer larvae survived at least 16 hr of heat shock at 30 or 32°C (*Figure 4—figure supplement 1A*). At 34°C, survival dropped dramatically after 12 hr, with no dependence on strain (*Figure 4—figure supplement 1A*). Finally, heat shock at 37°C could only be tolerated for 4 hr, again independent of the status of the GS shunt (*Figure 4—figure supplement 1A*). These results suggest that the GS is specifically involved in tolerance of desiccation, and possibly related stresses, in *C. elegans*.

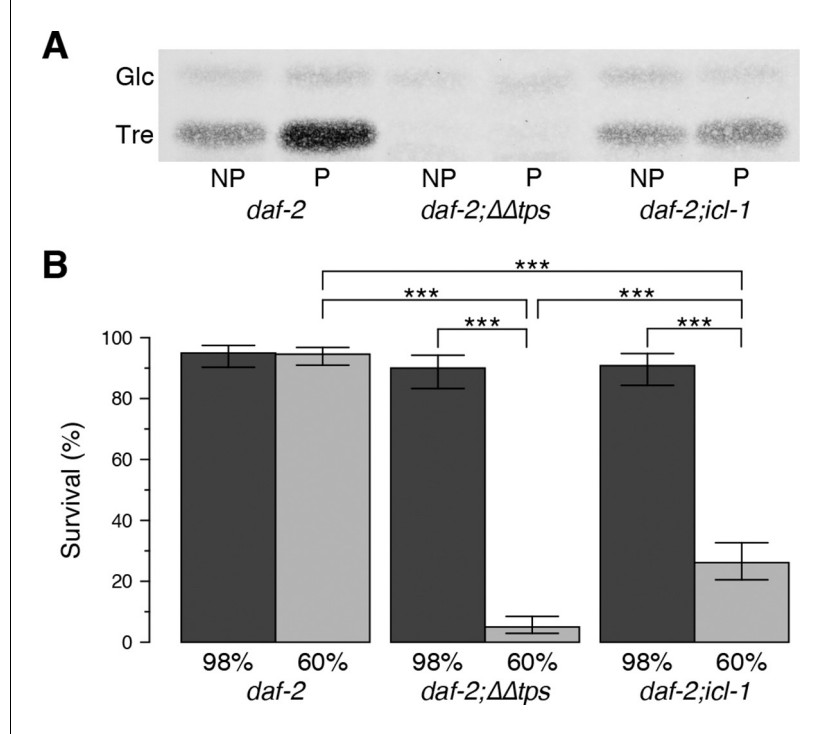

**Figure 4.** Effect of the glyoxylate shunt on desiccation tolerance. (**A**) Trehalose levels before (NP) and after (P) preconditioning in *daf-2, daf-2;ΔΔtps* and *daf-2;icl-1* dauer larvae after separation with HPTLC and visualization via Molisch's staining. Tre: Trehalose, Glc: Glucose. (**B**) Survival levels of the same strains at 98% and 60% RH after preconditioning (dark and light boxes, respectively). Statistical comparison was done with beta regression followed by multiple hypothesis testing. Analysis of deviance results indicate that survival levels depend both on the strain ($\chi_2^2$ = 124.64, p < 0.001) and the RH ($\chi_1^2$ = 141.46, p < 0.001). Error bars show 95% confidence intervals. *p < 0.001

The following figure supplement is available for figure 4:

**Figure supplement 1.** GS is not involved in heat-shock stress in *C. elegans* and *S. cerevisiae*.

## The glyoxylate pathway in *C. elegans* is mitochondrial

In plants, the GS takes place in a specialized peroxisome called the glyoxysome (*Eastmond and Graham, 2001*), whereas in yeast, GS enzymes are distributed between the cytosol and peroxisomes (*Duntze et al., 1969*; *Kunze et al., 2006*; *McCammon et al., 1990*). However, the localization of the GS pathway in nematodes has not been previously investigated. Using a bioinformatics tool (*Claros and Vincens, 1996*), we analyzed the *C. elegans* ICL-1 protein sequence. We identified a 20 amino acid N-terminal mitochondrial import sequence (MSSAAKNFYQVVKSAPKGRF) and calculated an 88% probability that the protein is imported into mitochondria. To determine the localization of ICL-1 (and thus the site of GS activity) in the worm, we generated a transgenic strain that expresses the ICL-1::GFP fusion protein under the control of the *icl-1* promoter, which mimics endogenous expression.

We first analyzed the localization of ICL-1::GFP in reproductive stage, actively feeding L3 larva. The protein was expressed at the highest levels in hypodermal cells (*Figure 5A*), although strong expression was also detected in the pharynx (*Figure 5—figure supplement 1A*) and gut (*Figure 5—figure supplement 1B*). In hypodermal syncytium, the protein was present in a tubular network interspersed with spherical structures (*Figure 5A*), resembling mitochondrial staining of *C. elegans* (*Lee et al., 2003*). To verify that ICL-1 is indeed localized to mitochondria, we fed worms the mitochondrial dye MitoTracker Red CMXRos (*Figure 5B*). In hypodermal cells

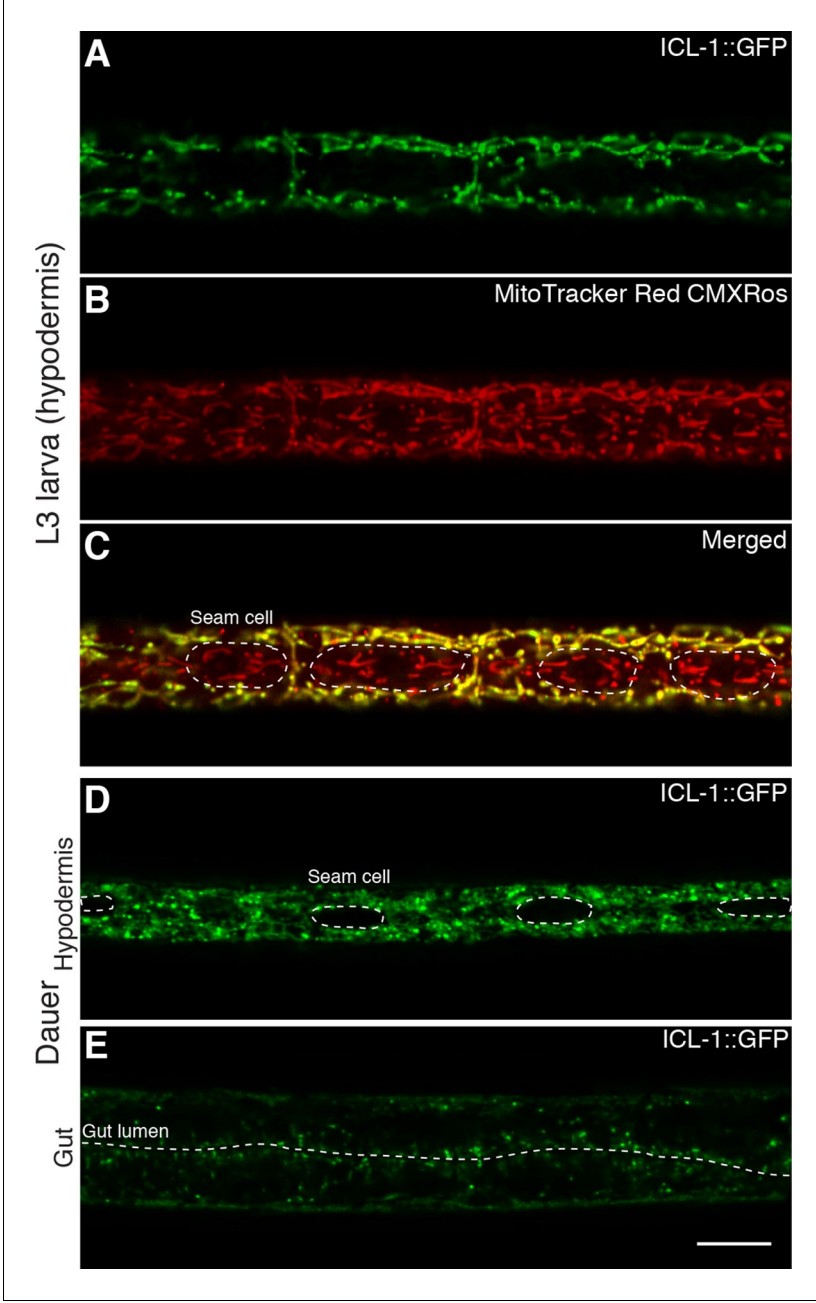

**Figure 5.** ICL-1 is a mitochondrial protein. (**A**) Subcellular localization of ICL-1::GFP in L3 hypodermis (**B**) Mitochondrial staining of L3 hypodermis. (**C**) Colocalization of mitochondria and ICL-1::GFP in L3 hypodermis. Seam cells are circled with dashed curves. (**D**) Subcellular localization of ICL-1::GFP in dauer hypodermis. Seam cells are circled with dashed curves. (**E**) Subcellular localization of ICL-1::GFP in dauer gut. Gut lumen is shown as a dashed line. Scale bar corresponds to 10 µm for all images.

The following figure supplement is available for figure 5:

**Figure supplement 1.** Expression of ICL-1 in different tissues.

expressing ICL-1, the MitoTracker and GFP signals fully overlapped, whereas seam cells did not express ICL-1 at all (*Figure 5C*). It should be noted that TPS-1, the key enzyme in trehalose biosynthesis, is almost exclusively localized to hypodermis and is not expressed in seam cells (*Penkov et al., 2015*).

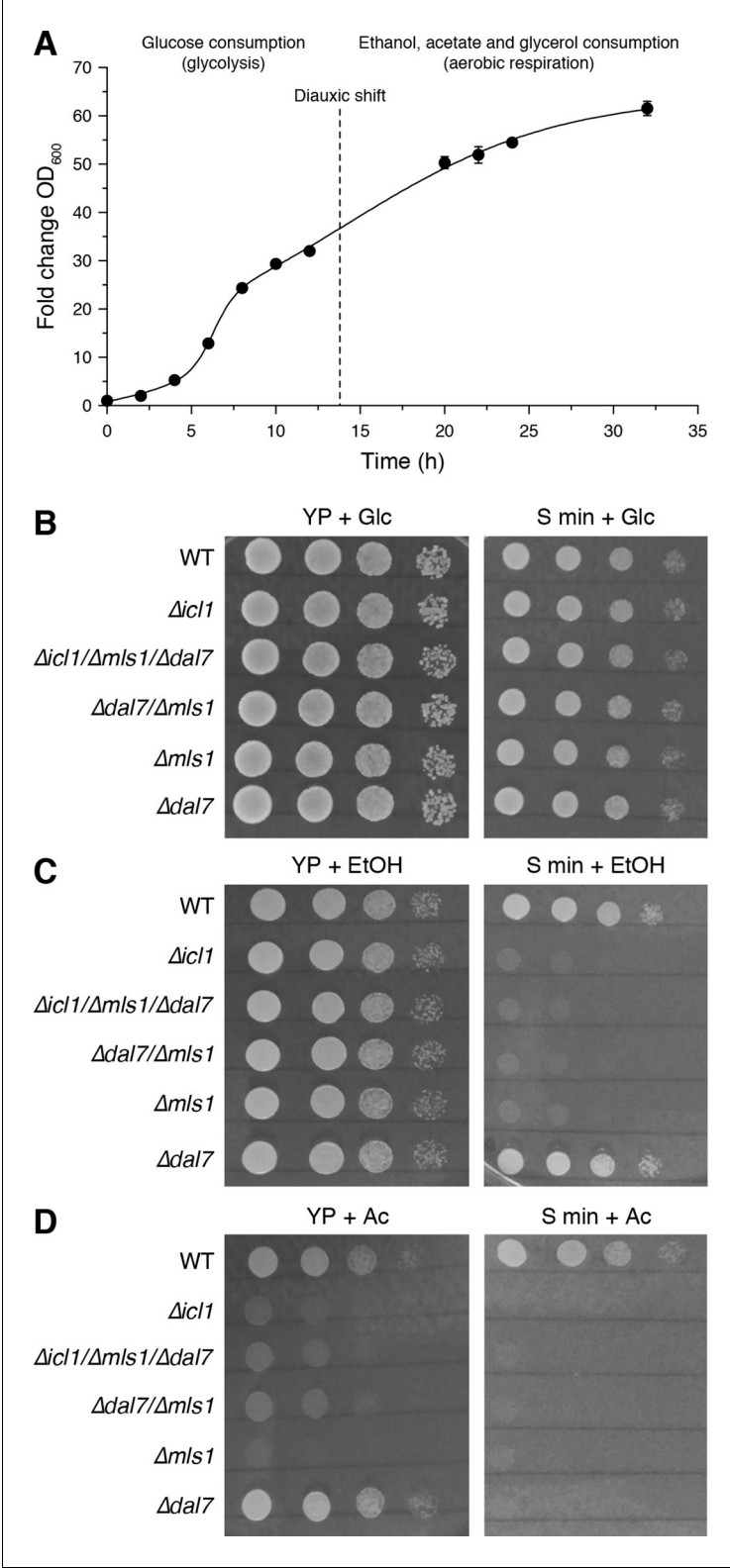

**Figure 6.** Growth of GS-deficient yeast cells in media with different carbon sources. (**A**) Growth of wild-type *S. cerevisiae* in YP + Glc medium in batch culture. The time of diauxic shift is shown with a dashed line. Error bars show standard deviation (n = 3). (**B–D**) Growth of wild-type (WT) or GS-deficient yeast in amino acid rich (YP) or minimal medium with glucose (**B**), ethanol (**C**) and acetate (**D**) as the carbon source. Note that particularly with

*Figure 6 continued on next page*

Figure 6 continued

acetate as the primary carbon source, GS mutants (Δicl1, Δmls1, Δdal7, Δmls1/Δdal7 and Δicl1/Δmls1/Δdal7) grow poorly regardless of amino acid availability.

Because they are non-feeding and impermeable, dauer larvae cannot be stained with Mito-Tracker. Nevertheless, at the subcellular level, the distribution of the GFP signal in dauer larvae closely resembled the mitochondrial network (*Figure 5D*). Once again, the hypodermis was the main tissue expressing ICL-1, although the protein was also expressed in the gut (*Figure 5E*). Collectively, these results indicate that *C. elegans* ICL-1 is mitochondrial, and suggest that in the worm, the GS occurs within or in association with mitochondria.

## The glyoxylate shunt is required for trehalose accumulation and desiccation tolerance in *S. cerevisiae*

Based on the striking conceptual similarity between *C. elegans* dauer larvae and *S. cerevisiae* cells entering stationary phase, we postulated the existence of a conserved mechanism for desiccation tolerance. As described earlier, dauer larvae accumulate large amounts of trehalose, despite the fact that worms in this stage of the life cycle do not feed or grow. A similar phenomenon occurs in budding yeast. In the presence of its preferred carbon source (glucose), yeast uses fermentative glycolysis during rapid proliferation (*Figure 1B*); under these conditions, very little trehalose accumulates (*Tapia and Koshland, 2014*). However, once glucose concentration falls, the cells undergo a diauxic shift, thereafter using aerobic respiration for their energetic needs in order to continue proliferation (*Figures 1D*, *6A*); eventually, as external energy sources are depleted, the cells enter stationary phase. Although both glycolytic and respiratory activities are low in stationary phase, the cells continue to accumulate trehalose and glycogen, which ultimately constitute >30% of total cell mass (*François and Parrou, 2001*; *Werner-Washburne et al., 1993*). Thus, yeast might also rely on alternate carbon metabolism to generate trehalose. Therefore, we asked whether the GS in yeast can be used to drive gluconeogenesis for synthesis of trehalose.

In stationary phase, yeast can consume ethanol, glycerol, and particularly acetate to generate acetyl-CoA, either directly or through gluconeogenesis. Acetyl-CoA can enter the TCA cycle as well as the GS (*Figure 1D*). In yeast, the GS is carried out by isocitrate lyase, Icl1p (*Fernandez et al., 1992*), and the malate synthases, primarily Mls1p but also Dal7p (*Hartig et al., 1992*). We first compared the growth rates of WT yeast harboring mutations in GS components (Δicl1, Δmls1, Δdal7, and combinations thereof) under conditions in which we altered carbon sources as well as the availability of free amino acids (*Figure 6B–D*).

As expected, GS mutants exhibited no significant growth defect when grown in high glucose, irrespective of the presence or absence of amino acids (*Figure 6B*). By contrast, when grown in high ethanol, GS mutants grew normally in YP medium, but their growth was impaired in minimal medium (S min) lacking free amino acids (*Figure 6C*). This suggests that free amino acids can feed into carbon consumption in the TCA cycle, as *C. elegans*, and that in the absence of free amino acids, the GS plays an important role in carbon metabolism. In addition, we compared the growth rates of WT and GS-deficient cells growing on acetate as the sole carbon source (*Figure 6D*). Under these conditions, yeast have high GS activity and exhibited elevated TCA-independent acetate metabolism (*Lee et al., 2011*; *Schweizer and Dickinson, 2004*). Regardless of the availability of free amino acids, GS mutants grew very poorly under these conditions (*Figure 6D*).

We predicted that after the diauxic shift, GS-deficient yeast would exhibit reduced synthesis and accumulation of trehalose and glycogen. To test this idea, we quantitated these metabolites in WT and GS-deficient cells (*Figure 7A–D*). In cells grown in glucose, following the diauxic shift (20 hr, black bars) and in stationary phase (48 hr, grey bars), WT cells accumulated considerable amounts of trehalose (*Figure 7A*) and glycogen (*Figure 7C*). By contrast, despite reaching high cell densities, GS-deficient cells contained low (but detectable) amounts of trehalose, but no detectable glycogen stores (*Figure 7A,C*) (*Figure 7—figure supplement 1A*). We also measured trehalose and glycogen in WT and GS-deficient cells grown with acetate as a carbon source. Under these conditions, we observed an even greater accumulation of trehalose and glycogen in WT cells, whereas GS-deficient cells contained low levels of trehalose and no detectable glycogen (*Figure 7B,D*). As controls, we

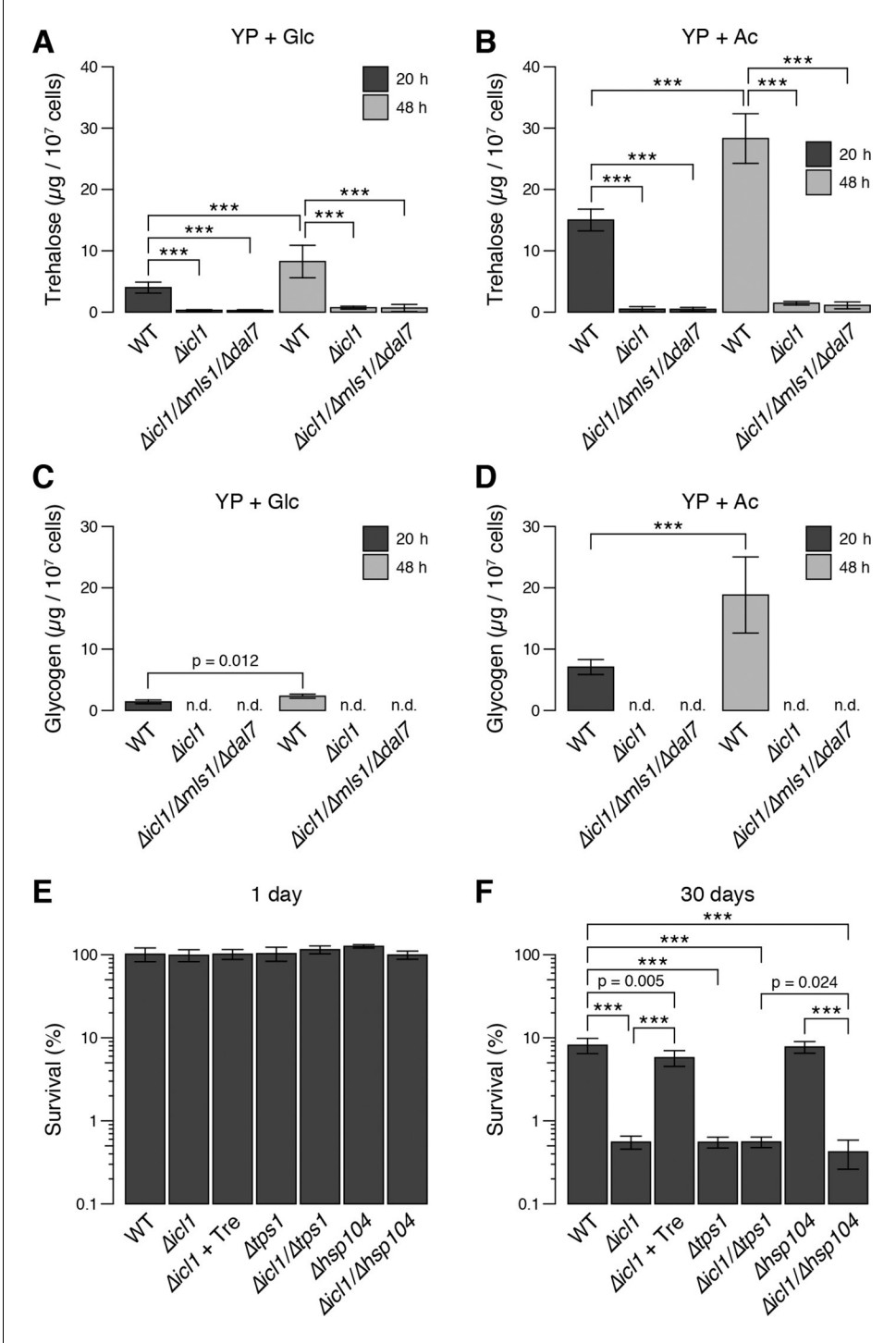

**Figure 7.** Trehalose/glycogen synthesis and desiccation tolerance in GS-deficient yeast cells. (A, B) Steady-state trehalose levels of WT, Δ*icl1* and Δ*icl-1/*Δ*mls1/*Δ*dal7* strains in YP + Glc (A) and YP + Ac (B) media after 20 hr (post-diauxic shift, dark bars) and 48 hr (stationary phase, light bars). (C, D) Steady-state glycogen levels under the same conditions. n.d.: Not detected/below assay sensitivity range. (E, F) Desiccation tolerance of the indicated WT and mutant yeast cells, measured after 24 hr (E) or 30 days (F) of desiccation. Error bars show 95% confidence intervals. ***p < 0.001.

The following figure supplement is available for figure 7:

**Figure supplement 1.** Cell density, trehalose and glycogen levels in WT and mutant yeast strains.

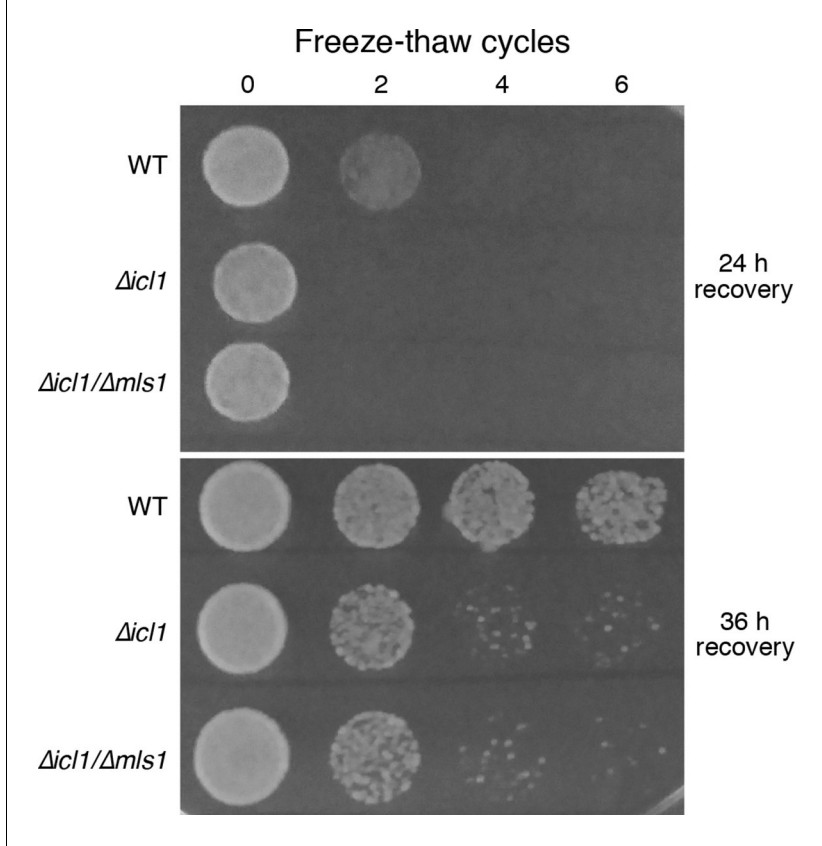

**Figure 8.** Resistance of WT and GS-deficient cells to freezing and thawing. Equal numbers of cells were subjected to multiple freeze-thaw cycles, and survival estimated by spotting onto YPD plates. The plates were imaged after different times of recovery, to more carefully observe survival phenotypes.

also measured the amounts of trehalose and glycogen in mutant cells lacking trehalose synthase (Tps1) or the heat-shock protein Hsp104, both of which are important for yeast desiccation tolerance (*Tapia and Koshland, 2014*). As expected, Δtps1 cells contained very low levels of trehalose, but high levels of glycogen, whereas Δhsp104 cells had no defects in trehalose (*Figure 7—figure supplement 1B*) or glycogen storage (*Figure 7—figure supplement 1C*). We also measured Tps1 and Tps2 amounts (*Figure 3—figure supplement 2B*), which were unchanged in wild type and GS-deficient cells. This rules out the trivial explanation that GS-deficient cells have limitations in trehalose 6-phosphate synthase levels, and therefore have lower trehalose amounts. Collectively, these data show that, much like preconditioned *C. elegans* dauer larvae, *S. cerevisiae* cells rely on TCA cycle-independent acetyl-CoA consumption through the GS to synthesize trehalose and glycogen.

Therefore, we asked whether this GS-dependent trehalose accumulation was required for desiccation tolerance in yeast. We grew saturated cultures of WT, GS-deficient, Tps1-deficient, or Hsp104 deficient cells, desiccated them for up to 30 days, and then rehydrated them. All strains survived very well after 1 day of desiccation (*Figure 7E*). The viability of cells in WT cultures remained high after 30 days of desiccation (*Figure 7F*). By contrast, Δicl1, Δicl1/Δtps1, and Δicl1/Δhsp104 cells exhibited very poor desiccation tolerance after 30 days, with viability at least 10-fold lower than that of WT cells and comparable to that of Δtps1 cells lacking trehalose altogether (*Figure 7F*). Importantly, Δicl1 cells supplemented with trehalose in the medium 24 hr before desiccation exhibited near-WT levels of desiccation tolerance (*Figure 7F*).

Finally, we expanded our study to investigate whether other environmental insults, such as heat-shock and freezing/thawing, were affected by GS deficiency. We first tested the ability of *S. cerevisiae* cells to tolerate elevated temperatures, subjecting wild type or GS-deficient mutants to heat shock at 50°C either at different cell densities, or for increasing amounts of time (*Figure 4—figure*

*supplement 1B and C*). Under both conditions, wild type as well as GS-deficient cells showed similar sensitivity to heat shock.

Another environmental insult that yeast seasonally encounter is freezing and thawing. This stress could conceivably affect cell membranes and proteins similarly to desiccation. We subjected stationary phase cultures of WT or GS-deficient cells (resuspended in water) to multiple freeze-thaw cycles, and monitored viability by simple spotting and growth assays (*Figure 8*). Although a considerable proportion of WT cells survived even after six freeze-thaw cycles, GS-deficient cells underwent a dramatic loss in viability after just two cycles (*Figure 8*).

Together, our observations demonstrate that the GS plays an essential role in yeast desiccation tolerance by promoting gluconeogenesis for the synthesis of trehalose. Our data also suggest that GS-dependent trehalose synthesis is critical for survival under conditions of other water-related stresses, such as freezing.

## Discussion

Here, we demonstrated that dauer larvae exist in a hypometabolic state in which metabolism is redirected largely towards gluconeogenesis. This state depends primarily on an active glyoxylate shunt (GS), which serves as the main route for synthesis of trehalose from TAG reserves during preconditioning. We also showed that budding yeast undergoes a conceptually convergent process. In order to survive desiccation, stationary phase yeast cells must produce high levels of trehalose from acetate or glycerol. This conversion can be successfully accomplished only in the presence of a functional GS.

In addition to the GS, gluconeogenesis/trehalose biosynthesis should use alternative carbon sources, such as amino acids, glycerol, and pyruvate, because *daf-2;icl-1* accumulated some trehalose upon preconditioning. Amino acids, for example, can be converted into their corresponding α-keto acids, which then undergo a specific series of reactions to enter the TCA cycle, in which they are subsequently used for gluconeogenesis (*Figure 1—figure supplement 1A*). However, even via utilization of amino acids or other metabolites, *daf-2;icl-1* dauer larvae cannot tolerate desiccation as well as *daf-2*. Our data indicate that an intermediate level of trehalose is insufficient for desiccation tolerance: Below some threshold level, trehalose cannot exert its protective effects and overcome the adverse consequences of desiccation. This hypothesis was previously explored in yeast (*Tapia et al., 2015*). Our results strongly suggest that a conceptually similar trehalose threshold may exist in the worm as well. It is worth to note that gluconeogenesis is in general associated with the energetic needs of the organism. In *C. elegans* and the yeast, however, it is additionally used as a defense against environmental stress. Taking advantage of the GS, worms and yeast devote large amounts of resources to the production of trehalose, which in turn protects the organism against desiccation or other water-related stresses, such as freezing and thawing during winter.

Although the GS was discovered and dissected at the molecular level almost 60 years ago, no clear physiological function has yet been assigned to it. For several decades, the germination of plant seeds was considered to be the most prominent process requiring this pathway, and it was assumed that production of sugars from seed oils was a prerequisite for germination. However, studies of *Arabidopsis* lines with no functional GS revealed that in the presence of light, this pathway is non-essential (*Eastmond et al., 2000*; *Eastmond and Graham, 2001*). Other studies suggested a requirement for the GS in fungal virulence (*Lorenz and Fink, 2001*), although at present we have no mechanistic understanding of why this would be the case. To the best of our knowledge, our study provides the first clearly defined physiological role for the GS in *C. elegans* and *S. cerevisiae*.

It remains unclear how the GS is regulated in animals. In the worm, regulation is directly connected to sensing of a desiccative environment (hygrosensation). We previously showed that hygrosensation is at least partially mediated by head neurons (*Erkut et al., 2013*). Although we still do not know the details of this sensation, one of its downstream effects is likely to be increased lipolysis. This process, followed by β-oxidation of fatty acids, yields acetyl-CoA, the fuel for GS. It is therefore reasonable to speculate that lipolysis determines the extent of the GS. Another finding that supports this view is the localization of ICL-1 within the organism. As shown above, ICL-1 is predominantly localized in the hypodermis. The major enzyme that synthesizes trehalose (TPS) is localized to the same tissue (*Penkov et al., 2015*), but is not expressed in the gut. On the other hand, the major TAG deposit in the form of fat droplets resides in the gut, and to some extent in the hypodermis

(*Mak, 2011*). Thus, synthesis of trehalose in the hypodermis must depend on the transport of fatty acids from the gut. Indeed, as we previously found, one of the fatty acid-binding proteins (FAR-3) is strongly upregulated at both the transcriptional and translational levels upon preconditioning ($\sim$160 and four fold, respectively) (*Erkut et al., 2013*). FAR-3 is predicted to have a 20 amino acid signal sequence for secretion, and could thus be involved in the transport of fatty acids between cells. These observations strongly suggest that the regulation of the GS by substrate availability is a complex process that depends on interactions between different tissues.

An interesting aspect of our study is that worm ICL-1 is localized to mitochondria, suggesting that the GS takes place in this organelle. By contrast, in plants and the yeast, this pathway is split between mitochondria and a specialized organelle (glyoxysome or peroxisome) or the cytoplasm. Furthermore, in contrast to yeast and many other organisms, *C. elegans* ICL-1 is a bifunctional enzyme with both glyoxylase and malate synthase activities. The physiological meaning of these differences remains elusive, but they suggest that transition into gluconeogenic mode is regulated differently in different organisms.

In summary, we showed that dauer larvae and stationary phase yeast switch to a gluconeogenic mode, in which the GS plays an essential role. In both species, loss of the GS is deleterious during desiccation. Our results reveal a novel physiological role for the GS and a conserved mechanism by which diverse organisms can regulate their metabolism to achieve desiccation tolerance.

## Materials and methods

### Worm strains and culture conditions

C. elegans wild-type (N2), *daf-2(e1370)III, icl-1(ok531)V, tps-1(ok373)X* and *tps-2(ok526)II* strains were received from *Caenorhabditis* Genetics Center (Minneapolis, MN), which is funded by the NIH Office of Research Infrastructure Programs (P40 OD010440). The glyoxylate shunt mutant *icl-1* was outcrossed twice with N2 and subsequently crossed to *daf-2* to generate *daf-2(e1370)III;icl-1(ok531) V (daf-2;icl-1)*. The trehalose-deficient strains *tps-2(ok526)II;tps-1(ok373)X (ΔΔtps)* and *tps-2(ok526)II; daf-2(e1370)III;tps-1(ok373)X (daf-2;ΔΔtps)* were previously generated in our group (*Penkov et al., 2010*).

The transgenic line was obtained by ballistic transformation of a fosmid construct encoding the C-terminal translational fusion protein ICL-1::eGFP, generated by our TransgeneOmics facility (*Sarov et al., 2012*). The construct was isolated and purified using a FosmidMAX DNA purification kit (Epicentre, Madison, WI) and sequenced to confirm its identity. Microparticle bombardment was performed as explained elsewhere (*Sarov et al., 2012*) . Transgenic worms showing the GFP marker and rescue of Unc phenotype were screened for 2 generations to pick up an integrated line. This strain was then outcrossed twice with N2 and finally crossed to *daf-2* to obtain *daf-2(e1370)III;Is[icl-1::GFP+unc-119] (daf-2;icl-1::GFP).*

Worms were maintained at 15°C on nematode growth medium (NGM) agar plates seeded with *Escherichia coli* NA22 (*Brenner, 1974*). Gravid adults on NGM agar plates were treated with alkaline hypochlorite solution (i.e., bleached) to purify eggs. Dauer larvae of Daf-c strains were obtained by growing these eggs in complete S medium (liquid culture) (*Sulston and Brenner, 1974*) at 25°C for 5 days unless stated otherwise. To obtain dauer larvae of other strains, we first let the eggs grow into gravid adults on sterol-depleted lophanol (4α-methyl-5α-cholestan-3β-ol)-substituted agarose plates at 20°C for 4 days (*Matyash et al., 2004*). Subsequently, these adults were bleached and their eggs were grown in cholesterol-free lophanol-substitued liquid culture at 25°C for 5 days. L3 larvae were obtained by growing eggs at 15°C in liquid culture for 3 days.

To radioactively label lipids and sugars in worms, we let the eggs grow on NGM agar plates supplemented with bacteria mixed with 10 μCi $^{14}$C-labeled sodium acetate (CH$_3$$^{14}$COONa, Hartmann Analytic, Germany) for 3 days at 15°C or 25°C until they became L3 or dauer larvae, respectively.

### Yeast strains and culture conditions

The prototrophic *Sacharomyces cerevisiae* CEN.PK strain background was used in all experiments (*van Dijken et al., 2000*). Strains that have been generated and used in this study are Δ*icl1* (MAT a Δ*icl1::NAT*), Δ*mls1* (MAT a Δ*mls1::KanMX*), Δ*dal7* (MAT a Δ*dal7::KanMX*), Δ*dal7/*Δ*mls1* (MAT a Δ*dal7::KanMX* Δ*mls1::Hyg*), Δ*icl1/*Δ*mls1/*Δ*dal7* (MAT a Δ*icl1::NAT* Δ*mls1::KanMX* Δ*dal7::Hyg*), Δ*tps1*

(MAT a Δtps1::Hyg), Δicl1/Δtps1 (MAT a Δicl1::NAT Δtps1:Hyg), Δhsp104 (MAT a Δhsp104::KanMX), Δicl1/Δhsp104 (MAT a Δicl1::NAT Δhsp104::KanMX), Tps1-FLAG (MAT a Tps1-FLAG::NAT), Tps2-FLAG (MAT a Tps2-FLAG::NAT), Δicl1/Tps1-FLAG (MAT a Δicl1::NAT Tps1-FLAG::Hyg), Δicl1/Tps2-FLAG (MAT a Δicl1::NAT Tps2-FLAG::Hyg), Δmls1/Tps1-FLAG (MAT a Δmls1::KanMX Tps1-FLAG::Hyg) and Δmls1/Tps2-FLAG (MAT a Δmls1::KanMX Tps2-FLAG::Hyg). Gene deletions were performed using standard PCR-based strategies (*Longtine et al., 1998*).

Standard formulations for rich medium (YP: yeast extract, peptone) or synthetic minimal medium (S: Yeast Nitrogen Base (YNB) and ammonium sulfate without amino acids) with the specified carbon source were used. The carbon sources were 2% dextrose, 2% ethanol + 2% glycerol or 2% sodium acetate. Cell growth in a specified medium was measured using a serial dilution assay on plates. Briefly, cells were grown in YP with 2% glucose for 12 hr, after which they were harvested, washed twice in water, and serial diluted in water (starting $OD_{600}$ = 1.0), following which, 5 μl drops were spotted onto agar plates containing YP or S medium with glucose, ethanol and glycerol, or acetate, and cell growth was measured by imaging the plates. Cell growth rates in YPD medium were measured by monitoring absorbance ($OD_{600}$) over time.

## C. elegans desiccation assay

Worms were harvested from liquid cultures or plates (after radioactive labeling) in distilled water and washed extensively to remove bacteria and debris. Preconditioning for subsequent biochemical analysis was done by first filtering dauer larvae on TETP membranes (Merck-Millipore, Germany) and then placing them in a controlled humidity chamber equilibrated at 98% RH (*Erkut et al., 2011*). After 4 days of incubation at 25°C, these worms were collected in distilled water and frozen. L3 or non-preconditioned dauer larvae were frozen right after they were harvested.

Desiccation survival assay was performed as described before (*Erkut et al., 2013*). Briefly, in duplicate, 5 μl of worm slurry (approximately 1000 worms) in distilled water was dropped into the middle of a 35 mm plastic dish and placed into a controlled humidity chamber equilibrated at 98% RH. After 4 days of preconditioning at 25°C, one replicate was transferred to another controlled humidity chamber equilibrated at 60% RH and kept there for 1 day at 25°C. Meanwhile, the other replicate was left in the 98% RH chamber. Finally, worms were rehydrated with distilled water for 2–3 hr at room temperature and transferred to NGM agar plates seeded with *E. coli*. They were let recover at 15°C overnight. Next day, alive and dead worms were counted to calculate the survival rate. This experiment was carried out on 3 different days with 3 technical replicates on each day for each treatment.

## S. cerevisiae desiccation assay

Desiccation tolerance assays were performed as described earlier (*Tapia and Koshland, 2014*), with slight modifications. Briefly, ∼$10^7$ cells were collected from batch cultures (grown for 96 hr in YPD), washed twice in dilute PBS, and brought to a final volume of 1 ml. Non-desiccated controls were plated on YPD agar for colony counting. Two hundred microliter aliquots were transferred to a 96-well tissue culture plate, centrifuged, and the excess water was removed. Cells were allowed to desiccate in a humid incubator at 27°C. Long-term desiccation experiments were kept for indicated time periods in a 96-well tissue culture plate at 27°C. Samples were resuspended in diluted PBS to a final volume of 200 μl, and plated for colony counting. The number of colony forming units per milliliter (cfu/ml) for each plate was measured, using an average from three independent controls. The relative viability of each experimental sample (done in biological triplicate) was determined by dividing the cfu/ml for that sample by the average cfu/ml of the control plates.

## C. elegans heat-stress survival assay

Worms were collected from liquid cultures and incubated at elevated temperatures for 4, 8, 12 or 16 hr. After each time point, worms were allowed to cool down at room temperature and survival rate was calculated after counting the survivors.

## S. cerevisiae heat-stress survival assay

S. *cerevisiae* strains were grown to stationary phase (72 hr) in YPD medium, after which cells were collected by centrifugation and washed twice with water. Subsequently, two different heat-stress

survival assays were performed. In the first one, cells were resuspended at decreasing cell densities, starting at an $OD_{600}$ of 1.0 and then serially diluted (1:10) up to an $OD_{600}$ of 0.001. These were subjected to severe heat shock at 50℃ for 30 min. 5 µl from each of these samples were spotted onto YPD plates. Cells were allowed to recover for ~30 hr before imaging the plates, and estimating survival. In the second assay, cells were resuspended at a single cell density ($OD_{600}$ of 0.1), and subjected to heat stress for 45, 60 and 75 min. 5 µl of each suspension was spotted onto YPD plates and cells were allowed to recover for ~30 hr before imaging and estimating survival.

### *S. cerevisiae* freeze-thaw assay

WT or GS-deficient *S. cerevisiae* strains were grown to stationary phase (72 hr) in YPD medium, after which cells were collected by centrifugation and washed twice with water. Subsequently cells were resuspended at at an $OD_{600}$ of 0.1. These were subjected to rapid freezing, followed by thawing at room temperature, for multiple cycles. 5 µl from each of these samples were spotted onto YPD plates. Cells were allowed to recover for the indicated times before imaging the plates, and estimating survival.

### Organic extraction

Worms were collected in 1 ml distilled water and homogenized by freezing in liquid nitrogen and subsequent thawing in a sonication bath for 5 times. The debris was pelleted by centrifugation at 25,000 g for 1 min at 4℃. A micro BCA assay kit (Thermo Fisher Scientific, Germany) was used to determine total soluble protein amounts from the supernatant. Next, the pellet was resuspended and the homogenate was extracted according to Bligh and Dyer's method (*Bligh and Dyer, 1959*). Briefly, homogenized sample in 1 ml water was mixed with 3.75 ml of chloroform–methanol (1:2, v/v) in glass tubes for at least 20 min. Then 1.25 ml of chloroform and 1.25 ml of water were added sequentially, with rigorous mixing after each addition. Phase separation was facilitated by centrifugation at 1,000 g for 15 min. Next, organic (lower) and aqueous (upper) phases were collected into fresh glass tubes using sterilized glass Pasteur pipettes. Organic fractions from radioactively labeled samples and all aqueous fractions were dried under vacuum with heating. Organic fractions from non-labeled samples were dried under nitrogen gas flow. All organic and aqueous fractions were dissolved in chloroform–methanol (2:1, v/v) and methanol–water (1:1, v/v), respectively. Non-labeled samples were normalized according to total soluble protein amounts measured from homogenates. For each mg of protein, organic and aqueous fractions were dissolved in 166 µl and 332 µl of the corresponding solvent, respectively. Labeled organic samples were dissolved in 100 µl of the corresponding solvent and total radioactivity in each sample was measured by a scintillation counter.

### Trehalose measurement from worm extracts

After sample homogenization and protein measurement prior to organic extraction, trehalose measurement was performed in some samples using a trehalose assay kit with a modified protocol (Megazyme, Ireland). First, 40 µl of each homogenate supernatant was heated at 95℃ for 5 min to inactivate endogenous enzymes. Next, reducing sugars in the homogenate were reduced to sugar alcohols by adding 40 µl of freshly prepared alkaline borohydride (10 mg/ml sodium borohydride in 50 mM sodium hydroxide) into each tube and incubating at 40℃ for 30 min with shaking at 300 rpm. Then, the mixture was neutralized by adding 100 µl of 200 mM acetic acid. Subsequently, the pH was adjusted by adding 40 µl of 2 M imidazole buffer (pH 7.0). 70 µl of the final mixture was transferred to a plastic cuvette and the reaction mixture was added (70 µl of 2 M imidazole buffer, 35 µl of $NADP^+$/ATP mix, 7 µl of hexokinase/glucose 6-phosphate dehydrogenase mix and 700 µl of distilled water). The reaction was carried out at room temperature for 15 min. Then the basal absorbance at 340 nm was measured ($A_1$). After that, 7 µl of trehalase was added and incubated for 15 min at room temperature before the final absorbance at 340 nm was measured ($A_2$). Trehalose concentration was calculated from the difference of absorbance values and normalized to the protein amounts measured from the same samples. This experiment was carried out on 3 different days with 3 technical replicates on each day for each treatment. Median values of technical replicates were used for calculations.

## Trehalose measurement from yeast samples

Trehalose and glycogen from yeast samples were quantified as described previously, with minor modifications (*Shi et al., 2010*). Cell samples were collected and pelleted. Cell pellets were quickly washed with 1 ml of ice-cold water and then resuspended in 0.25 ml of 0.25 M sodium carbonate and stored at -80°C until processed. For batch cultures, 20 $OD_{600}$ total cells were collected. After resuspension in water, 0.5 ml of cell suspension was transferred to two capped Eppendorf tubes (one tube for glycogen assay and the other tube for trehalose assay). When sample collections were complete, cell samples (in 0.25 M sodium carbonate) were boiled at 95–98°C for 4 hr, and then 0.15 ml of 1 M acetic acid and 0.6 ml of 0.2 M sodium acetate were added into each sample. Each sample was incubated overnight with 1 U/ml amyloglucosidase (Sigma-Aldrich, India) rotating at 57°C for the glycogen assay, or 0.025 U/ml trehalase (Sigma-Aldrich, India) at 37°C for the trehalose assay. Samples were then assayed for glucose using a glucose assay kit (Sigma-Aldrich, India). Glucose assays were done using a 96-well plate format. Samples were added into each well with appropriate dilution within the dynamic range of the assay (20–80 µg/ml glucose). The total volume of sample (with or without dilution) in each well was 40 µl. The plate was pre-incubated at 37°C for 5 min, and then 80 µl of the assay reagent from the kit was added into each well to start the colorimetric reaction. After 30 min of incubation at 37°C, 80 µl of 12 N sulfuric acid was added to stop the reaction. Absorbance at 540 nm was determined to assess the quantity of glucose liberated from either glycogen or trehalose.

## TPS activity assay in *C. elegans*

TPS activity assay was based on *Dmitryjuk et al. (2014)*. Worm homogenates were prepared in 1 ml of 0.9% NaCl (w/v). Total soluble protein and the initial amount of trehalose were measured as described above. Reaction was carried out with 100 µl of the lysate in 40 mM acid-ammonia buffer (pH 4.2), 2 mM $MgCl_2$, 0.2 mM UDP-glucose, 0.2 mM glucose 6-phosphate and 80 µM trehalase inhibitor at 37°C for 30 min in a total volume of 0.5 ml. Subsequently, the resulting trehalose 6-phosphate was dephosphorylated with 1 U of alkaline phosphatase in 100 mM phosphoric buffer (pH 8) at 37°C for 30 min. Reaction was stopped via heating the samples to 95°C for 5 min. Next, the final amount of trehalose was measured and the difference from initial level of trehalose was calculated. This difference was then normalized to the total soluble protein amount. The unit enzyme activity is defined as the normalized molar amount of trehalose 6-phosphate produced in 1 min.

## Detection of Tps1 and Tps2 proteins in *S. cerevisiae*

A 3X-FLAG epitope tag was added to the carboxy termini of Tps1 and Tps2 at the endogenous chromosomal locus in the indicated strains. Tps-FLAG containing wild type and GS-deficient strains were grown to stationary phase (72 hrs) in YPD medium, cells were harvested by centrifugation, and proteins were extracted by first precipitating with 10% trichloroacetic acid (TCA), followed by removal of TCA, and solubilization of the protein extracts in SDS-glycerol sample buffer, normalizing for total protein. Proteins were separated on an SDS-PAGE gel and detected with standard immunoblotting for the FLAG epitope, using a mouse anti-FLAG antibody, and HRP-conjugated rabbit anti-mouse IgG secondary antibody.

## Thin-layer chromatography

High-performance thin-layer chromatography (HPTLC) was used to separate and visualize molecules of interest. The TLC system was developed using non-radioactive amino acid and sugar standards, based on *Tweeddale et al. (1998)*. Individual amino acid samples were first separated on 1 dimension for either mobile phases, visualized by ninhydrin staining, and their corresponding $R_f$ values were calculated. Then they were mixed and separated on 2 dimensions. Individual $R_f$ values calculated from the 1D TLC runs coincided largely with each molecule in question also on 2D. Furthermore, the positions of glutamate and glutamine were confirmed in another set of experiments, where glutamate- or glutamine-lacking mixtures of amino acids were separated on 2D. Localization of sugars on the 2D TLC system was done similarly, only using Molisch staining as the visualization method.

Before any analysis, sample normalization following organic extraction was confirmed by loading 5 µl of each organic fraction on an HPTLC plate (Merck, Germany), eluting with chloroform–

methanol–water (45:18:3, v/v/v) and visualizing with copper acetate solution (3% copper acetate and 8% ortho-phosphoric acid in water) after baking at 200°C. It was expected that the phospholipids should have comparable levels in every sample.

Sugars were separated using chloroform–methanol–water (4:4:1, v/v/v) as the mobile phase and visualized with Molisch's reagent (3% 1-naphtol in sulfuric acid (96%)–water–ethanol (13:8:80, v/v/v)) after baking at 180°C. For comparison of trehalose levels of dauer larvae before and after preconditioning, 8 µl of aqueous fraction was used.

Triacylglycerol levels were compared by running 4 µl of the organic fraction with the solvent system petroleum ether (b.p. 60–80°C)–diethyl ether–glacial acetic acid (82:18:1, v/v/v) and visualizing with copper acetate.

We used the 2-dimensional TLC approach to compare the metabolites in aqueous fractions of radioactively labeled worm extracts. An amount of aqueous fraction equivalent to 2700 worms (L3 or dauer larvae) was applied as a spot to an HPTLC plate and eluted on the first dimension with 1-propanol–methanol–ammonia (32%)–water (28:8:7:7, v/v/v/v). Then, the plate was dried for 15 min and eluted on the second dimension with 1-butanol–acetone–glacial acetic acid–water (35:35:7:23, v/v/v/v). Finally, it was sprayed with EN[3]HANCE spray surface autoradiography enhancer (Perkin Elmer, Waltham, MA) and exposed on an X-ray film, which was developed by standard methods.

Differences in the amounts of radioactively labeled trehalose were calculated via densitometry. For this purpose, TLCs were exposed to X-ray films for a shorter period (2 hr), which was optimized to prevent the saturation of spots. Subsequently, films were developed and scanned, after which trehalose spot intensities were calculated using Fiji (fiji.sc) software.

## Mitochondrial staining and microscopy

*daf-2:icl-1::GFP* L3 and dauer larvae were grown in liquid culture for 3 days at 15 and 25°C, respectively. Subsequently, they were pelleted in a 15 ml tube and resuspended in 100 µl of the original culture medium still containing bacteria. MitoTracker Red CMXRos (Thermo Fisher Scientific, Germany) stock solution (1 mM in DMSO) was diluted into 10 µM in the worm-bacteria suspension. Our previous experience shows that worms can tolerate up to 1% DMSO in the medium, therefore 1:100 dilution of the MitoTracker solution is the highest concentration that can be safely achieved. Worms were incubated in this solution for 1.5 hr at room temperature in dark. Next, the excess dye was washed off with M9 buffer and worms were resuspended again in 100 µl. They were incubated in M9 for 30 min so that they could defecate the excess dye in the gut. Finally, L3 and dauer larvae were anesthetized in 20 mM and 50 mM sodium azide, respectively.

Meanwhile, agarose pads on microscope slides were prepared and 5 µl of worm suspension was transferred on them. They were immediately covered with a coverslip and sealed with nail polish. We used a Zeiss LSM 700 inverted laser scanning confocal microscope and a Zeiss LCI Plan-Neofluar 63×/1.3 Imm Corr DIC M27 objective to image mitochondria (Zeiss, Germany). Simultaneously, GFP was excited at 488 nm and the emission below 550 nm was acquired by the first PMT while MitoTracker Red CMXRos was excited at 555 nm and the emission above 560 nm was acquired by the second PMT. Optical sections at $0.1 \times 0.1 \times 0.5$ µm$^3$ x-y-z resolution were collected in a 4D hyperstack. Final images were adjusted for intensity and merged in Fiji. No non-linear adjustments were done.

## Oxygen consumption assay

Oxygen consumption of worms was measured with a Seahorse XF$^e$96 system (Seahorse Bioscience, North Billerica, MA). L3 and dauer larvae of *daf-2* and *daf-2;icl-1* strains were grown in liquid culture for 3 days. Then, they were collected and washed extensively to remove excess bacteria and debris. Approximately 100 worms were pipetted into each well of a 96-well Seahorse XF$^e$ assay plate, except for the 4 corner wells, which were used to estimate the background measurement. Initial oxygen consumption rate (OCR) was measured until the readings stabilized. Then, sodium azide with phenol red indicator was injected into half of the wells at a final concentration of 20 mM. The rest of the wells were injected only phenol red indicator in water. OCR was measured once again until it stabilized. Then 4 subsequent measurements were done in 6.5 min intervals. Finally, exact number of worms in each well were counted and used for normalization. A small number of abnormal readings

were also filtered out at this stage. On average, 7–8 wells (technical replicates) were used for each condition.

Normalized OCR values were averaged for the last 4 measurements for each strain, stage and injection. OCR after water injection was named as total OCR (tOCR) and OCR after azide injection was named as non-mitochondrial OCR (nmOCR). The difference of tOCR and nmOCR was calculated for each time point (measurement) and named as mitochondrial OCR (mOCR) or respiration rate.

## Bioinformatics analysis

Biochemical pathway analysis was done by querying the KEGG database for *C. elegans* proteins using NemaPath software (www.nematode.net) (*Wylie et al., 2008*). Data acquired from these queries were cross referenced to our previously published microarray data (*Erkut et al., 2013*). Sequences of worm proteins were obtained from WormBase (www.wormbase.org). ICL-1 sequences were submitted to MitoProt (ihg.gsf.de/ihg/mitoprot.html) to predict the probability of mitochondrial import of ICL-1 (*Claros and Vincens, 1996*). Signal sequence for FAR-3 was predicted with SignalP 4.0 (www.cbs.dtu.dk/services/SignalP-4.0) (*Petersen et al., 2011*).

## Statistical analysis

All statistical analyses were done in R environment (www.r-project.org). Trehalose and glycogen levels, as well as OCRs were compared with analysis of variance (ANOVA) followed by Tukey's honestly significant differences (HSD) post-hoc test. Trehalose/glycogen amounts were log-transformed prior to model fit, normality was confirmed with QQ-plots and Shapiro-Wilk test, homoscedasticity with Levene's test. Survival rates after desiccation and rehydration were compared with beta regression as described before (*Erkut et al., 2013*), followed by Type II analysis of deviance for generalized linear models. Prior to beta regression, fit to beta distribution was confirmed with QQ-plots. Statistical power was calculated via power analysis when possible. The maximum Type I error rate was set as $\alpha$ = 0.05 for all tests. Data are presented as mean ± standard error for *C. elegans* trehalose levels and survival rates, and mean ± 95% confidence limit for other measurements unless stated otherwise.

# Acknowledgements

We would like to thank Hans-Joachim Knölker for the synthesis of lophanol, Mihail Sarov and Elisabeth Loester for the *icl-1::GFP* fusion construct, Susanne Ernst for the ballistic transformation of the construct, Davide Accardi for technical support with confocal microscopy, and Sider Penkov for acquiring data on the induction of $^{14}$C-labeled trehalose.

# Additional information

## Funding

| Funder | Grant reference number | Author |
|---|---|---|
| Max Planck Institute of Molecular Cell Biology and Genetics | Internal funds | Teymuras V Kurzchalia |
| Wellcome Trust-DBT India Alliance | Intermediate Fellowship | Sunil Laxman |

The funders had no role in study design, data collection and interpretation, or the decision to submit the work for publication.

## Author contributions

CE, Respiration rate measurement, TLC analysis of metabolites, Quantification of trehalose, Desiccation survival assay, Subcellular localization of ICL-1 in C. elegans, Conception and design, Acquisition of data, Analysis and interpretation of data, Drafting or revising the article; VRG, Quantification of trehalose, Analysis of 14C-labeled trehalose induction, TPS activity assay and heat-shock assay in C. elegans, Acquisition of data, Analysis and interpretation of data, Drafting or revising the article; SL, Growth assays, desiccation, freeze-thaw and heat-shock survival assays, Measurement of Tps1 and Tps2 levels, and quantification of trehalose as well as glycogen levels in S. cerevisiae, Conception

and design, Acquisition of data, Analysis and interpretation of data, Drafting or revising the article; TVK, Conception and design, Analysis and interpretation of data, Drafting or revising the article

## Author ORCIDs

Cihan Erkut, http://orcid.org/0000-0003-4378-9075
Teymuras V Kurzchalia, http://orcid.org/0000-0002-1683-8460

## Additional files

### Major datasets

The following previously published dataset was used:

| Author(s) | Year | Dataset title | Dataset URL | Database, license, and accessibility information |
|---|---|---|---|---|
| Erkut C, Vasilj A, Boland S, Habermann B, Shevchenko A, Kurzchalia TV | 2013 | Transcription profiling by array of C. elegans dauer stage larve with or without extreme dessication stress to study the nematode's molecular strategies in surviving extreme desiccation | http://www.ebi.ac.uk/arrayexpress/experiments/E-MEXP-3899/ | Publicly available at ArrayExpress Archive of Functional Genomics Data (accession no. E-MEXP-3899) |

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
