## [Decision Letter]

Thank you for submitting your work entitled "The glyoxylate shunt is essential for desiccation tolerance in *C. elegans* and the budding yeast" for consideration by *eLife*. Your article has been reviewed by two peer reviewers, and the evaluation has been overseen by a Reviewing Editor and Richard Losick as the Senior Editor.

The reviewers have discussed the reviews with one another and the Reviewing Editor has drafted this decision to help you prepare a revised submission.

While there is interest in this work and the observations are potentially exciting, it was felt that the work presented is too preliminary as the reviewers noted in a number of spots that the observations were of low quality and/or lacked any effort at quantification. There was also a general feeling that there must be some effort to assess the impact of this phenomenon in other stages of *C. elegans* to justify the specificity of the major claim. The following are specific detailed major revisions that coincide with the reviewers’ concerns.

1) Trehalose content should be quantified in all experiments using appropriate replicated experimental design and proper statistical analysis. The current presentations are considered to be insufficiently empirical.

2) Similarly enzyme levels for Tps1 and Tps2 should be quantified if the claim is to focus on the substrate level accumulation.

3) There was also concern about if the effect is truly specific to the dauer state and an attempt should be made to test other states for stress resistance to test the specific argument being made.

*Reviewer #1:*

The article by Erkut et al. convincingly provides a role for the glyoxylate shunt in desiccation tolerance in *C. elegans* and budding yeast. The experiments carried out are convincing and the use of the two model systems rather than the normal one is a considerable boon to the manuscript. Whilst I am find the story compelling there are two issues concerning the radioblots which I think could yet further improve the manuscript. Firstly, it is not clearly defined how the spots were identified – I assume co-running with authentic standards? However, this should be clearly stated. Secondly, whilst the differences are massive and I thus do not doubt the authors’ interpretation could they quantify them? Also is this replicated enough times to perform statistical comparison? As mentioned I do not doubt the findings but it would be nice to define the amount of trehalose production in the glyoxylate mutant. These modifications should be relatively easy to address.

The authors pass over the fact that the two enzymes in *C. elegans* are encoded by a single protein. I find this fascinating and worthy of further comment even though I realise this would only be speculative.

*Reviewer #2:*

The manuscript of Erkut et al. provides compelling evidence that the glyoxylate shunt is critical for their survival of dauer *C. elegans* and stationary cultures of budding yeast to desiccation because the shunt contributes to the accumulation of trehalose. The steps of the glyoxylate shunt and the underlying enzymology were described many decades ago but the biological function of this metabolic pathway remained elusive. Thus the manuscript provides potentially an interesting new insight into the importance of regulating metabolism to achieve the proper characteristics of the quiescent state. Many experiments presented in the manuscript are rigorously performed and the quality of the data is excellent. However, the authors need to address important issues to strengthen the manuscript.

The authors assume that the glyoxylate mutants impact trehalose accumulation by impacting the level of the substrates of the trehalose biosynthetic enzymes. However it is possible that the altered metabolic state reduces the level of the biosynthetic enzymes themselves. The authors should quantitate the levels of Tps1 and Tps2 in *icl* mutants in yeast and worms. This concern raises the more general question whether the function of the glyoxylate pathway is specific to desiccation tolerance or whether it contributes to the overall metabolic state of the dauer or stationary cells. The authors should test whether other representative characteristics of these states, survival, heat tolerance, etc. are altered in the glyoxylate shunt mutants. The answers to these questions might alter the authors' conclusions and discussion.

The Discussion is mostly a rehash of the Introduction and Results, missing many possibilities to make their results more appealing to a general audience. The authors fail to put their results in context of what is known about metabolic pathways and cell states. For example, have any other metabolic pathways been associated with specific cell states? They also fail to put their discovery into context about what is known about the regulation of trehalose biosynthesis.

---

## [Author Response]

While there is interest in this work and the observations are potentially exciting, it was felt that the work presented is too preliminary as the reviewers noted in a number of spots that the observations were of low quality and/or lacked any effort at quantification. There was also a general feeling that there must be some effort to assess the impact of this phenomenon in other stages of C. elegans to justify the specificity of the major claim. The following are specific detailed major revisions that coincide with the reviewers’ concerns.

1) Trehalose content should be quantified in all experiments using appropriate replicated experimental design and proper statistical analysis. The current presentations are considered to be insufficiently empirical.

In all experiments concerning trehalose, its content is quantified and mean levels are statistically compared. At least three biological replicates and 2-3 technical replicates were used in these quantifications (subsection “Organic extraction”).

2) Similarly enzyme levels for Tps1 and Tps2 should be quantified if the claim is to focus on the substrate level accumulation.

Although, in response to the comment from Reviewer 2, we no longer strongly claim substrate level regulation, we nevertheless measured TPS enzyme levels in the worm as well as in the yeast (subsection “Trehalose measurement from yeast examples”, Figure 3—figure supplement 2). In worm, we quantified the total trehalose-6-phosphate synthase (TPS) activity in lysates of *daf-2* and *daf-2;icl-1*. We think this approach is very informative, because the activity of an enzyme may depend on other factors such as post-translational modifications or inhibitors. In yeast we measured Tps1 and Tps2 protein levels by expressing FLAG-tagged proteins. In both organisms TPS levels in wild type and glyoxalate shunt deficient mutants are similar.

3) There was also concern about if the effect is truly specific to the dauer state and an attempt should be made to test other states for stress resistance to test the specific argument being made.

We already investigated other states for desiccation tolerance (Erkut et al. 2011, Current Biology) and showed that only the dauer larva can survive desiccation. Other developmental stages are sensitive to even slight decreases in ambient humidity. Therefore, none of the assays we present in this work could have been performed in a reproductive stage worm.

Reviewer #1:

The article by Erkut et al. convincingly provides a role for the glyoxylate shunt in desiccation tolerance in C. elegans and budding yeast. The experiments carried out are convincing and the use of the two model systems rather than the normal one is a considerable boon to the manuscript. Whilst I am find the story compelling there are two issues concerning the radioblots which I think could yet further improve the manuscript.

Firstly, it is not clearly defined how the spots were identified – I assume co-running with authentic standards? However, this should be clearly stated.

The reviewer correctly points out that the details of mapping metabolites on 2D TLC system are not clearly explained in the methods. The procedure is now described in detail in the revised manuscript as the following:

“The TLC system was developed using non-radioactive amino acid and sugar standards, based on Tweeddale et al., 1998. Individual amino acid samples were first separated on 1 dimension for either mobile phases, visualized by ninhydrin staining, and their corresponding R_f_ values were calculated. Then they were mixed and separated on 2 dimensions. Individual R_f_ values calculated from the 1D TLC runs coincided largely with each molecule in question also on 2D. Furthermore, the positions of glutamate and glutamine were confirmed in another set of experiments, where glutamate- or glutamine-lacking mixtures of amino acids were separated on 2D. Localization of sugars on the 2D TLC system was done similarly, only using Molisch staining as the visualization method.”

Secondly, whilst the differences are massive and I thus do not doubt the authors’ interpretation could they quantify them? Also is this replicated enough times to perform statistical comparison? As mentioned I do not doubt the findings but it would be nice to define the amount of trehalose production in the glyoxylate mutant. These modifications should be relatively easy to address.

This is a very important point. In the initial manuscript, we quantified the total (non-radioactive) trehalose levels from biological triplicates in *daf-2* and *daf-2;icl-1* dauer larvae. In the dauer larva before preconditioning, steady-state trehalose levels are approximately 100 µg trehalose per mg total soluble protein. This is increased 5 fold upon preconditioning in *daf-2* larvae, but only 2 fold in *daf-2; icl-1* larvae. Both increases are statistically significant. During the revision of the manuscript, upon the advice of the reviewer, we also measured the fold change of trehalose labeled with ^[13]^C-acetate in these strains (subsection “Thin-layer chromatography”, Figure 3). Because we could not measure the absolute amounts of labeled trehalose, we could only measure the difference in folds. The induction of labeled trehalose is ~6 fold for *daf-2* and 2 fold for *daf-2;icl-1* (subsection “An intact GS is required for utilization of acetate/fatty acids for trehalose biosynthesis”), suggesting that the utilization of acetate (and thus fatty acids) for gluconeogenesis and trehalose biosynthesis depends on the existence of a functional GS. All measurements were done with 3 biological replicates and 2 technical replicates.

The authors pass over the fact that the two enzymes in C. elegans are encoded by a single protein. I find this fascinating and worthy of further comment even though I realise this would only be speculative.

Indeed, it is extremely interesting that in the worm, ICL-1 has two separate domains for both isocitrate lyase and malate synthase activities. Already after cloning of this enzyme in Epstein’s group 20 years ago, it raised great excitement and it was even compared to the operon logic in bacteria (Liu, F. et al. Dev Biol 169, 399–414). We briefly discuss about this in the revised manuscript (Discussion section).

*Reviewer #2:*

The manuscript of Erkut et al. provides compelling evidence that the glyoxylate shunt is critical for their survival of dauer C. elegans and stationary cultures of budding yeast to desiccation because the shunt contributes to the accumulation of trehalose. The steps of the glyoxylate shunt and the underlying enzymology were described many decades ago but the biological function of this metabolic pathway remained elusive. Thus the manuscript provides potentially an interesting new insight into the importance of regulating metabolism to achieve the proper characteristics of the quiescent state. Many experiments presented in the manuscript are rigorously performed and the quality of the data is excellent. However, the authors need to address important issues to strengthen the manuscript.

*The authors assume that the glyoxylate mutants impact trehalose accumulation by impacting the level of the substrates of the trehalose biosynthetic enzymes. However it is possible that the altered metabolic state reduces the level of the biosynthetic enzymes themselves. The authors should quantitate the levels of Tps1 and Tps2 in icl mutants in yeast and worms.*

Thank you for pointing out this possibility. We measured enzyme levels in the worm as well as in the yeast (subsection “Trehalose measurement from worm extracts”, Figure 3—figure supplement 2). In worm, we quantified the total trehalose 6-phosphate synthase (TPS) activity in lysates of *daf-2* and *daf-2;icl-1* (Figure 3—figure supplement 2). We think this approach is very informative, because the activity of an enzyme may depend on other factors such as post-translational modifications or inhibitors. In yeast we measured Tps1 and Tps2 protein levels by expressing FLAG-tagged proteins under the control of their own promoters (Figure 3—figure supplement 2). In both organisms TPS levels in wild type and glyoxalate shunt deficient mutants are similar. These results rule out the trivial explanation that GS-deficient worms/cells have limitations in trehalose biosynthesis and therefore have lower levels of trehalose.

This concern raises the more general question whether the function of the glyoxylate pathway is specific to desiccation tolerance or whether it contributes to the overall metabolic state of the dauer or stationary cells. The authors should test whether other representative characteristics of these states, survival, heat tolerance, etc. are altered in the glyoxylate shunt mutants. The answers to these questions might alter the authors' conclusions and discussion.

As requested by the reviewer, we tested how responses to other stress conditions (i.e., heat-shock in *C. elegans* and *S. cerevisiae* and freeze-thaw stress in *S. cerevisiae*) depend on the glyoxalate shunt (subsections “*C. elegans* heat-stress survival assey” and “*S. cerevisiae* heat-stress survival assay”, Figure 4—figure supplement 1, Figure 8). Our results indicate that the sensitivity of both organisms to heat shock is independent of the GS. Remarkably, however, GS-deficient yeast cells were very sensitive to freeze-thaw stress compared to wild type cells. These results collective strengthen our conclusion that the GS is specifically involved in water-related stress tolerance, such as desiccation or freezing.

The Discussion is mostly a rehash of the Introduction and Results, missing many possibilities to make their results more appealing to a general audience. The authors fail to put their results in context of what is known about metabolic pathways and cell states. For example, have any other metabolic pathways been associated with specific cell states? They also fail to put their discovery into context about what is known about the regulation of trehalose biosynthesis.

The Discussion was rewritten and significantly shortened by about 450 words. Now it is more focused on our novel findings, putting them in a more general context in terms of global regulation of metabolism.